# Presynaptic Calcium Channels

**DOI:** 10.3390/ijms20092217

**Published:** 2019-05-06

**Authors:** Sumiko Mochida

**Affiliations:** Department of Physiology, Tokyo Medical University, Tokyo 160-8402, Japan; mochida@tokyo-med.ac.jp; Tel.: +81-333516141

**Keywords:** Ca^2+^ channels, synaptic transmission, G-proteins, synaptic proteins, Ca^2+^ binding proteins

## Abstract

Presynaptic Ca^2+^ entry occurs through voltage-gated Ca^2+^ (Ca_V_) channels which are activated by membrane depolarization. Depolarization accompanies neuronal firing and elevation of Ca^2+^ triggers neurotransmitter release from synaptic vesicles. For synchronization of efficient neurotransmitter release, synaptic vesicles are targeted by presynaptic Ca^2+^ channels forming a large signaling complex in the active zone. The presynaptic Ca_V_2 channel gene family (comprising Ca_V_2.1, Ca_V_2.2, and Ca_V_2.3 isoforms) encode the pore-forming α1 subunit. The cytoplasmic regions are responsible for channel modulation by interacting with regulatory proteins. This article overviews modulation of the activity of Ca_V_2.1 and Ca_V_2.2 channels in the control of synaptic strength and presynaptic plasticity.

## 1. Introduction

Presynaptic Ca^2+^ entry into the active zone (AZ) occurs through voltage-gated Ca^2+^ (Ca_V_) channels which are activated membrane depolarization and triggers synchronous neurotransmitter release from synaptic vesicles (SVs). Multiple mechanisms regulate the function of presynaptic Ca^2+^ channels [1,2,3,4]. The channel activity for opening, closing, or inactivation in response to membrane depolarization changes every few milliseconds during and after neuronal firing, resulting in control of synaptic strength [3,4]. Following a brief overview of Ca^2+^ channel structure/function, this article reviews the molecular and cellular mechanisms that modulate the activity of presynaptic Ca^2+^ channels in the regulation of neurotransmitter release and in the induction of short-term synaptic plasticity. To understand the physiological role of Ca^2+^ channel modulation in the regulation of synaptic transmission, a model synapse formed between sympathetic, superior cervical ganglion (SCG) neurons in culture was employed for functional study of channel interaction with G proteins, SNARE proteins, and Ca^2+^-binding proteins which sense residual Ca^2+^ in the AZ after the arrival of an action potential (AP).

## 2. Presynaptic Ca^2+^ Channels

Ca^2+^ currents have diverse physiological roles and different pharmacological properties. Early investigations revealed distinct classes of Ca^2+^ currents which were identified with an alphabetical nomenclature [5]. P/Q-type, N-type, and R-type Ca^2+^ currents are observed primarily in neurons, require strong depolarization for activation [6], and are blocked by specific polypeptide toxins from snail and spider venoms [7]. P/Q-type and N-type Ca^2+^ currents initiate neurotransmitter release at most fast synapses [1,8,9]. The Ca^2+^ channels are composed of four or five distinct subunits (Figure 1a) [8,10]. The α1 subunit incorporates the conduction pore, the voltage sensors and gating apparatus, and target sites of toxins and intracellular regulators. The α1 subunit is composed of about 2000 amino acid residues and is organized in four homologous domains (I–IV) (Figure 1b). Each domain consists of six transmembrane α helices (S1 through S6) and a membrane-associated P loop between S5 and S6. The S1 through S4 segments serve as the voltage sensor module, whereas transmembrane segments S5 and S6 in each domain and the P loop between them form the pore module [11]. The intracellular segments serve as a signaling platform for Ca^2+^-dependent regulation of neurotransmission, as discussed below.

Ca^2+^ channel α1 subunits are encoded by ten distinct genes in mammals, which are divided into three subfamilies by sequence similarity [2,8,13]. The Ca_V_2 subfamily members Ca_V_2.1, Ca_V_2.2, and Ca_V_2.3 channels conduct P/Q-type, N-type, and R-type Ca^2+^ currents, respectively [2,8,9,13].

Ca_V_ channels are complexes of a pore-forming α1 subunit and auxiliary subunits. Skeletal muscle Ca_V_ channels have three distinct auxiliary protein subunits [8] (Figure 1a), the intracellular β subunit, the disulfide-linked α2δ subunit complex, and the γ subunit having four transmembrane segments. In contrast, brain neuron Ca_V_2 channels are composed of the pore-forming α1 and the auxiliary β subunit [14]. The auxiliary subunits of Ca^2+^ channels have an important influence on their function [15,16]. The Ca_V_β subunit shifts their kinetics and voltage dependence of activation and inactivation [15,16]. Cell surface expression of the α1 subunits is enhanced by the Ca_V_β subunit [15,16]. The α2δ subunits are potent modulators of synaptic transmission. The α2δ subunits increase not only Ca_v_1.2 but also Ca_v_2.2, Ca_v_2.1 currents, suggesting that the α2δ subunits enhance trafficking of the Ca_V_ channel complex [17]. Expression of α2δ subunits also appears to play a role in setting release probability [18]. Further details of these regulatory interactions are discussed below.

## 3. Intracellular Molecules Modulate Presynaptic Ca^2+^ Channels Activity

### 3.1. G Proteins

Presynaptic Ca^2+^ currents are reduced in magnitude by activation of G protein-coupled receptors for neurotransmitters at nerve terminals [19,20]. Gβγ subunits released from heterotrimeric G proteins of the Gi/Go class [19,20] bind directly to α1 subunits of the N-type Ca^2+^ channel [21,22] at the N terminus [23], the intracellular loop connecting domains I and II [21,24], and at the C terminus [25] (Figure 1b). Gβγ causes a positive shift in the voltage dependence of activation of the Ca^2+^ current [26,27,28]. The Gβγ-induced reduction of Ca^2+^ currents can be reversed by strong positive depolarization [26,27,28]. Reversal of this inhibition by depolarization provides a point of intersection between chemical and electrical signal transduction at the synapse and can potentially provide novel forms of short-term synaptic plasticity that do not rely on residual Ca^2+^.

The subtype of Ca_V_β can influence the extent and kinetics of Gβγ mediated inhibition and this regulation also depends on the subtype of Gβ involved [29,30]. Gβγ interacts with multiple sites on the N-terminus, I–II linker, and the C-terminus of the α1 subunit. Binding of Gβγ causes a conformational shift that promotes interaction of the N-terminus “inhibitory module” with the initial one-third of the I–II-linker. Strong membrane depolarization leads to unbinding of Gβγ and loss of interaction between the N-terminus and the I–II linker. This depends upon binding of Ca_V_β subunit to the α interaction domain (AID) on the I–II linker. In the absence of Ca_V_β1 subunit binding with tryptophan mutation in the AID (W391) of the Ca_V_2.2 α1 subunit, Ca^2+^ channel inhibition still occurs but cannot be reversed by strong depolarization. Ca_V_β2a, that is palmitoylated at two N-terminal cysteine residues, can still bind to the α1 subunit and permit voltage-dependent relief of the inhibition [31]. It is possible that binding of Ca_V_β1 to the AID induces a rigid α-helical link with domain IS6, and this transmits the movement of the voltage-sensor and activation gate to the I–II linker to alter the Gβγ binding pocket at depolarized potentials [32].

Specific Gβ subunits have been shown to be responsible for the Ca_V_2 channel modulation in different neurons. In rat SCG neurons Ca_V_2.2 channels are differentially modulated by different types of Gβ subunits, with Gβ_1_ and Gβ_2_ being most effective, Gβ_5_ showing weaker modulation, and Gβ_3_ and Gβ_4_ being ineffective [33,34,35]. In contrast, in rat stellate ganglion neurons, Gβ_2_ and Gβ_4_ but not Gβ_1_ subunit are responsible for the coupling of Ca_V_2.2 channels with noradrenaline receptors [36]. In the transfected human embryonic kidney tsA-201 cell line, Ca_V_2.2 channel inhibition, with Gβ_1_ and Gβ_3_ being more effective than Gβ_4_ and Gβ_2_, and no significant modulation being induced by Gβ_5_ [37]. Gβ subunit-induced inhibition of Ca_V_2.1 channel differed from those observed with the Ca_V_2.2 channel. Ca_V_2.1 channels exhibited more rapid rates of recovery from inhibition than those observed with Ca_V_2.2 channels, on average, twice as rapidly for the Ca_V_2.1 channels, indicating that Gβ binding to this channel subtype is less stable [37].

Regulation of the Ca_V_2.2 channels also involves the interplay between Ca^2+^ channels and G protein interaction. Syntaxin-1A, a presynaptic plasma membrane protein, is required for G protein inhibition of presynaptic Ca^2+^ channels [38]. Physical interaction between syntaxin-1A and Ca^2+^ channels is a prerequisite for tonic Gβγ modulation of Ca_V_2.2 channels, suggesting that syntaxin-1A mediates a colocalization of Gβγ subunits and Ca_V_2.2 channels, thus resulting in a more effective G protein coupling to, and regulation of, the channel. The interactions between syntaxin, G proteins, and Ca_V_2.2 channels are part of the structural specialization of the presynaptic terminal [39].

G proteins also induce voltage-independent inhibition of Ca_V_2 channels through intracellular signaling pathways [1,19,40]. This often involves the Gq family of G proteins, which regulate the levels of phosphatidylinositide lipids by inducing hydrolysis of phosphatidylinositol bisphosphate via activation of phospholipase C enzymes [41]. Acetylcholine release from rat sympathetic neurons is reduced through this pathway via presynaptic muscarinic receptors activation [42].

### 3.2. Active Zone Proteins

Rab-interacting molecule (RIM), an AZ protein required for SVs docking and priming [43,44,45,46,47,48], and synaptic plasticity [49], interacts with the C-terminal cytoplasmic tails of Ca_V_2.1 and Ca_V_2.2 channels [46,48,50,51] (Figure 1b). The interaction is essential for recruiting Ca^2+^ channels to the presynaptic AZ [46] and determines channel density and SVs docking at the presynaptic AZ [48]. RIM-binding proteins, RIM-BPs, also interact with Ca_V_2.1 and Ca_V_2.2 channels [51], and are selectively required for high-fidelity coupling of AP-induced Ca^2+^ influx to Ca^2+^-stimulated SVs exocytosis [52]. The tripartite complex of RIM, RIM-BPs, and C-terminal tails of the Ca_V_2 channels regulate the recruitment of Ca_V_2 channels to AZs. Interaction of RIM with Ca_V_β subunits shifts the voltage dependence of inactivation to more positive membrane potentials, increasing Ca^2+^ channel activity [53]. In contrast, Ca_V_β subunits interaction with CAST/ERC2 shifts the voltage dependence of activation to more negative membrane potentials [54]. Positive regulation of presynaptic Ca^2+^ channel activity by RIM and CAST/ERC2, in addition to their function in SVs docking, increase the release probability of SVs docked close to Ca_V_2 channels. Furthermore, Munc13, required for SVs priming, controls Ca_V_2 channels shortly after AP firing to guarantee transmitter release for continuous neural activity [55].

### 3.3. t-SNAREs and Synaptotagmin-1

SV (v)-SNARE synaptobrevin 2 and presynaptic plasma membrane (t)-SNAREs syntaxin-1 and SNAP-25 are required for fusion of SVs with a plasma membrane to release neurotransmitters [56]. Both Ca_V_2.1 and Ca_V_2.2 channels at the presynaptic nerve terminals colocalize densely with syntaxin-1A [57,58,59], and also form a complex of with SNARE proteins [60,61,62] dependent on Ca^2+^ with maximal binding at 20 μM and reduced binding at lower or higher concentrations of Ca^2+^ [63]. The t-SNARE proteins syntaxin-1A and SNAP-25, but not the v-SNARE synaptobrevin, bind to the intracellular loop between domains II and III of the α_1_ subunit of Ca_V_2.2 (amino acid residues 718-963) named as the synprint site (Figure 1b) [64,65]. Ca_V_2.1 channels have an analogous synprint site, and different channel isoforms have distinct interactions with syntaxin and SNAP-25 [66,67], suggesting specialized regulatory properties for synaptic modulation.

t-SNAREs interacting with presynaptic Ca_V_2.1 and Ca_V_2.2 channels regulate channel activity (Figure 3a). Syntaxin-1A or SNAP-25 shifts the voltage dependence of inactivation toward more negative membrane potentials and reduces the availability of the channels to open [68,69,70]. Coexpression of SNAP-25 can reverse the inhibitory effects of syntaxin-1A [69,71]. The transmembrane region of syntaxin-1A and only a short segment within the H3 helix are critical for channel modulation [72], whereas the synprint site binds to the entire H3 helix in the cytoplasmic domain of syntaxin-1A [63,64,72]. Deletion of the synprint site weakened the modulation of the channels by syntaxin-1A, but did not abolish it, arguing that the synprint site acts as an anchor in facilitating channel modulation but is not required absolutely for modulatory action.

Dependent on Ca^2+^ concentration, syntaxin-1 interacts with either the synprint site or synaptotagmin-1; at low Ca^2+^ concentrations, syntaxin-1 binds synprint, while at higher concentrations (>30 μM) it associates with synaptotagmin-1 [63]. Synaptotagmin-1, -2, and -9 serve as the Ca^2+^ sensors for the fast, synchronous neurotransmitter release [56,73,74]. The Ca^2+^ binding site C2B domain of synaptotagmin-1 interacts with the synprint sites of both Ca_V_2.1 and Ca_V_2.2 channels (Figure 1b) [75]. Synaptotagmin-1 can relieve the inhibitory effects of SNAP-25 on Ca_V_2.1 channels [70,76]. Relief of Ca^2+^ channel inhibition by the formation of the synaptotagmin/SNARE complex favors Ca^2+^ influx. This is a potential mechanism to increase the release probability of SVs docked close to Ca_V_2 channels [4].

Interaction of syntaxin-1A and SNAP-25 with the synprint site is controlled by phosphorylation of the synprint site with protein kinase C (PKC) (Figure 1b) [65] and Ca^2+^/calmodulin-dependent protein kinase II (CaMKII) [77]. The negative shift of steady-state inactivation of Ca_V_2.2 channels caused by syntaxin is blocked by PKC phosphorylation [65,71]. Thus, phosphorylation of the synprint site may serve as a biochemical switch controlling the SNARE-synprint interaction.

### 3.4. Ca^2+^-Sensor Proteins

Ca^2+^ elevation regulates Ca_V_2.1 channels activity by its binding to CaM [8,78,79,80,81] and related neuron-specific Ca^2+^-binding proteins, CaBP1, VILIP-2 [82,83,84], and NCS-1 (frequenin) [85]. The presynaptic Ca_V_2.1 channel proteins consist of a pore-forming α_1_ subunit associated with β, and possibly α_2_δ subunits (Figure 1a) [86]. The intracellular C terminus of the α1 subunit [81] called the IQ-like motif, which begins with the sequence isoleucine-methionine (IM) instead of isoleucine-glutamine (IQ), and the nearby downstream CaM-binding domain (CBD) are the interacting sites with these Ca^2+^-binding proteins (Figure 1b). Displacement with alanine in the IQ-like domain inhibited Ca^2+^-dependent Ca_V_2.1 channels facilitation [78,81], whereas deletion of CBD inhibited Ca^2+^-dependent Ca_V_2.1 channels inactivation [79,80,81,83,84]. Ca^2+^/CaM-dependent inactivation of Ca_V_2.1 channels, dependent on global elevations of Ca^2+^, is observed in transfected cells overexpressing Ca_V_2.1 channels [78,79,80] and in the nerve terminals of the calyx of Held [87,88] where Ca_V_2.1 channels are densely localized. In contrast, the large neuronal cell bodies of Purkinje neurons [89] or SCG neurons [90] rarely show Ca^2+^-dependent Ca_V_2.1 channels inactivation.

## 4. Negative Regulation of Neurotransmitter Release by Gβγ protein/Ca_V_2 Channel Complex

Receptor-activated Gβγ modulation of presynaptic Ca_V_2 channels is a potent negative regulation of neurotransmitter release. Electrophysiological recordings of Ca^2+^ currents and synaptic transmission at the calyx of Held demonstrated this type of negative regulation by activation of GABA-B receptors or metabotropic glutamate receptors [91,92]. Optical measurements of Ca^2+^ transients at the nerve terminals of the parallel fibers of cerebellar granule cells innervating Purkinje neurons has also demonstrated similar modulation by activation of CB1 receptors [93]. This Gβγ-mediated inhibition of Ca^2+^ channels is relieved by depolarization. At autapses formed by single hippocampal pyramidal neurons, trains of AP-like stimuli relieve the inhibition of synaptic transmission caused by activation of GABA-B receptors, resulted in facilitation of synaptic transmission, which was blocked by inhibition of Ca_V_2.1 channels with neurotoxins [94]. Thus, presynaptic firing could reverse the neurotransmitter-mediated G protein inhibition of synaptic transmission. Regulator of G protein signaling-2 (RGS-2), which speeds GTPase activity of the α subunit of the activated G protein α-GTP, determines short-term plasticity in hippocampal neurons by regulating Gi/o-mediated inhibition of presynaptic Ca^2+^ channels. RGS-2 relieves the inhibition, resulting in a higher basal probability of release and synaptic facilitation [95]. However, at parallel fibers synapses onto Purkinje cells, this form of facilitation is not responsible for short-term synaptic plasticity [96].

In SCG neurons noradrenaline shortens AP duration by reducing Ca^2+^ entry through Ca_V_2.2 channels, resulting in a reduction of transmitter release [97]. Purified Gβγ microinjected into presynaptic SCG neurons in culture reduced synaptic transmission, and the Gβγ introduced neurons caused no further reduction of synaptic transmission with noradrenaline [97]. Thus, Gβγ is a potent negative regulator of neurotransmission inhibiting presynaptic Ca_V_2.2 channels activity. The α1 subunit contains several Gβγ interaction sites, including the amino-terminal (NT) and I–II loop (Figure 1b). The “NT peptide” and an I–II loop α interaction domain “AID peptide” microinjected into presynaptic SCG neurons under long-term culture attenuated noradrenaline-induced G protein modulation (Figure 2) and inhibited synaptic transmission [98]. In acutely dissociated SCG neurons, NT and AID peptides reduced whole-cell Ba^2+^ current amplitude, modified voltage dependence of Ca^2+^ channel activation, and attenuated noradrenaline-induced G protein modulation (Figure 2) [98]. Co-application of NT and AID peptide negated inhibitory actions. Furthermore, a mutation within NT abolished inhibitory effects of the NT peptide [98]. Effects of Ca_V_2.2 channel peptides demonstrate that the Ca_V_2.2 amino-terminal and I–II loop serve as molecular determinants for Ca^2+^ channel function to inhibit synaptic transmission and to attenuate G protein modulation.

## 5. Synchronous Neurotransmitter Release Regulated by Ca^2+^ Channel/SNARE Proteins Complex

Synprint peptides derived from Ca_V_2.2 channels reduced transmitter release from the microinjected presynaptic SCG neurons in culture, due to competitive uncoupling of the endogenous Ca^2+^ channel-SNARE proteins interaction in nerve terminals [99]. Synprint peptides selectively inhibited fast synchronous synaptic transmission, while they increased late asynchronous release (Figure 3b). Similarly, synprint peptides reduced transmitter release from embryonic *Xenopus* spinal neurons [100]. Increasing the external Ca^2+^ concentration effectively rescued this inhibition, implying that synprint peptides competitively displaces docked SVs away from Ca^2+^ channels, and this effect can be overcome by increasing Ca^2+^ influx into presynaptic terminals [100].

At the calyx of Held, presynaptic neurons express P/Q-, N- and R-type Ca^2+^ currents in postnatal day 7 rats. P/Q-type Ca^2+^ currents are more effective than N-type Ca^2+^ currents and R-type Ca^2+^ currents in eliciting neurotransmitter release [101,102,103]. The high efficiency of P/Q-type Ca^2+^ currents to initiate neurotransmitter release is correlated with the close localization of Ca_V_2.1 channels near docked SVs [104], as shown by immunocytochemistry [105], suggesting localization of Ca_V_2 channels determines the efficiency of neurotransmitter release in response to neural activity.

Ca_V_2 channels interaction with SNARE proteins, that is dependent on Ca^2+^ concentration [63], have two opposing effects: at the pre-firing state synaptic transmission is blocked by enhancing Ca_V_2 channels inactivation, whereas immediately after AP firing tethering SVs near the point of Ca^2+^ entry enhances synaptic transmission. The overexpression of a syntaxin mutant that is unable to regulate Ca_V_2.2 channels, but still binds to them [72], increased the efficiency of synaptic transmission at Xenopus neuromuscular junctions, as reflected in increased quantal content [106]. In contrast, injected synprint peptides reduced the basal efficiency of synaptic transmission, as reflected in reduced quantal content of synaptic transmission [106]. These results demonstrate a bidirectional regulation of synaptic transmission in vivo by interactions of Ca_V_2.2 channels with SNARE proteins.

## 6. Presynaptic Plasticity Induced by Ca^2+^-Sensors-Mediated Ca_V_2.1 Channel Modulation

At most fast synapse in the central nervous system, Ca_V_2 channels are expressed diversely. In contrast, synaptic transmission of long-term cultured sympathetic SCG neurons, forming a well-characterized cholinergic synapse [107,108], is mediated by Ca_V_2.2 channels [109,110]. The physiological role of presynaptic Ca_V_2.1 channel modulation by Ca^2+^-sensors was explored by exogenously expressed α1 subunit derived from the brain Ca_V_2.1 channel that functionally generates P/Q type currents with other endogenous subunits in SCG neuron [111]. Section 6 describes presynaptic plasticity induced by modulation of the Ca_V_2.1 channel that is mediated by CaM or expression of neuron-specific Ca^2+^-sensor proteins, monitoring excitatory postsynaptic potentials (EPSPs) evoked by various patterns of presynaptic APs firing in the presence of the blocker of endogenous Ca_V_2.2 channels [109].

### 6.1. Ca^2+^/CaM Mediates Synaptic Depression and Facilitation

Modulation of presynaptic Ca^2+^ channels has a powerful influence on synaptic transmission [90]. The cytoplasmic regions of the α1 subunit are the target of regulatory proteins for channel modulation (Figure 1B). Brain-derived α1 subunit of the Ca_V_2.1 channel mediates transmitter release from the transfected SCG neurons [111]. The transmitter release changes after AP firing due to modulation of Ca_V_2.1 channel interacting with Ca^2+^ bound CaM (Figure 4) [90]. CaM has two Ca^2+^ binding sites, N and C robes. The N-robe sensing rapid and higher increase in Ca^2+^ concentration [112] initiates synaptic depression, and following facilitation is mediated by the C-robe sensing lower Ca^2+^ concentration. EPSPs recorded by pairs of APs with varied stimulation intervals show paired-pulse depression (PPD) and facilitation (PPF) (Figure 4a). PPD with a short interval (<50 ms) was blocked by deletion of the CBD, while PPF with intermediate interval (50–100 ms) was blocked by mutation of the IQ-like motif. Thus, the decline in Ca^2+^ elevation after the first AP causes temporal regulation of the Ca_V_2.1 channel interacting with CaM, resulting in a change in the transmitter release efficacy (Figure 4b). The time-dependent opposing modulation of the Ca_V_2.1 channel activity may support a stable synaptic transmission.

Neural information in vivo is encoded in bursts of AP firing. Short-term presynaptic plasticity caused by APs bursts involves the CaM-dependent regulation of Ca_V_2.1 channel. Mutation of the IQ-like motif potentiated reduction of the release efficacy, whereas the deletion of the CBD increased the release efficacy (Figure 4c, IM-AA/ΔCBD). Thus, during APs bursts, CaM binding to the CBD controls negatively the release efficacy, whereas CaM binding to IQ-like motif controls it positively. At a higher frequency of APs burst over 20 Hz, the release efficacy of SCG neurons mediated by Ca_V_2.1 channels reduced gradually (Figure 4c, WT), suggesting that the CaM-dependent inactivation of Ca_V_2.1 channels shapes the time course of short-term synaptic plasticity by determining the timing of the peak of synaptic facilitation during APs bursts as well as the steady-state level of synaptic depression at the end of the APs bursts.

### 6.2. Neuron-Specific Ca^2+^-Sensor Proteins Mediate Synaptic Depression and Facilitation

CaBP1, VILIP-2, and NCS-1 are members of a subfamily of neuron-specific Ca^2+^-sensor proteins (nCaS) that possess four EF-hand Ca^2+^-binding motifs. CaBP-1, VILIP-2, and NCS-1 bind to the same site as CaM, and modulate Ca_V_2.1 channel activity. CaBP1, highly expressed in the brain and retina [114], causes rapid inactivation of Ca_V_2.1 channels, binding to the CBD [84]. VILIP-2, highly expressed in the neocortex and hippocampus [115], increases Ca^2+^-dependent facilitation of Ca_V_2.1 channels but inhibits Ca^2+^-dependent inactivation of Ca_V_2.1 channels, binding to both IQ-like motif and CBD [83]. NCS-1, the classical example of facilitation of synaptic activity by nCaS, reduces Ca^2+^-dependent inactivation of P/Q-type Ca^2+^ currents through interaction with the IQ-like motif and CBD without affecting peak current or activation kinetics [85].

Synaptic transmission of SCG neurons transfected with CaBP1 and VILIP-2 changed by their modulation of Ca_V_2.1 channels with binding residual Ca^2+^ [113]. APs burst at 10 Hz induces synaptic facilitation followed by synaptic depression due to endogenous CaM. CaBP1 coexpressed with Ca_V_2.1 channels, significantly reduced the synaptic facilitation and enhanced the synaptic depression (Figure 4d) [113]. In contrast, VILIP-2 coexpressed with Ca_V_2.1 reduced the synaptic depression and enhanced the synaptic facilitation (Figure 4d) [113]. CaBP1 and VILIP-2 have opposing effects on short-term synaptic plasticity, either favoring synaptic depression or facilitation, suggesting that nCaS via regulation of presynaptic Ca^2+^ channels may play a critical role in determining the diversity of short-term synaptic plasticity at CNS synapses.

The expression of NCS-1 in presynaptic SCG neurons does not affect synaptic transmission, eliminating effects of this nCaS on endogenous N-type Ca^2+^ currents [85]. However, in SCG neurons expressing Ca_V_2.1 channels, coexpression of NCS-1 induces facilitation of synaptic transmission in response to paired APs and trains of APs, and this effect is lost in Ca_V_2.1 channels with mutations in the IQ-like motif and CBD [85]. These results reveal that NCS-1 directly modulates Ca_V_2.1 channels to induce short-term synaptic facilitation, and further demonstrate that nCaS are crucial in fine-tuning short-term synaptic plasticity.

### 6.3. Temporal Regulation of Release Efficacy by Ca^2+^-Sensor Proteins

The opening of Ca^2+^ channel creates a steep gradient of Ca^2+^ elevation in the AZ, where each nCaS has a different affinity and binding speed to Ca^2+^ [112]. The affinity is CaM (5–10 μM) > CaBP1 (2.5 μM) >VILIP-2 (~1 μM) [116]. CaM has a lower affinity and a higher binding speed to Ca^2+^ than nCaS, suggesting a temporal regulation of Ca_V_2.1 channel activity by CaM versus nCaS. Their affinity and binding speed to Ca^2+^ determinate timing of the Ca_V_2.1 channel modulation. Thus differential effects of CaM and nCaS on facilitation and inactivation of the presynaptic Ca_V_2.1 channels would substantially change the encoding of the synaptic properties in response to bursts of APs firing [117].

Time window of the CaM- and nCaS-induced Ca_V_2.1 channel modulation after AP firing can be estimated by the paired-pulse protocol applying to SCG neurons transfected with Ca_V_2.1 channels. CaM mediated PPD with a short interval (<100 ms), and PPF with intermediate interval (20–100 ms) (Figure 2a). In contrast, NCS-1 induced PPF with a shorter interval (30–50 ms) [85]. CaBP1 induced PPD with interval <150 ms, while VILIP-2 induced PPF with an interval of 50–250 ms [113]. These data suggest that CaM modulates Ca_V_2.1 channels shortly after Ca^2+^ entry and lasts 100 ms, while NCS-1 acts much shorter and CaBP1 and VILIP-2 actions last longer than CaM effects. The time-dependent action of CaM and nCaS reflects the decline rate of Ca^2+^ concentration at the Ca_V_2.1 channels after an AP firing. The divergent actions of CaM and nCaS on Ca_V_2.1 channels fine-tune the function and regulatory properties of presynaptic P/Q-type Ca^2+^ currents, allowing a greater range of input-output relationships and causing various short-term plasticity at different synapses [4].

### 6.4. CaMKII Saves as Effector Checkpoint for Ca^2+^ Entry

CaMKII is the most prominent Ca^2+^/CaM-dependent regulator of postsynaptic response [118,119,120,121] and presynaptic function [122,123,124,125]. The autophosphorylated form of CaMKII [7], which does not require the catalytic activity of the enzyme [126], binds to the α1 subunit of Ca_V_2.1 channels upstream of the IQ-like motif, and enhances the activity by slowing inactivation and positively shifting the voltage dependence of inactivation [126]. The dephosphorylation of CaMKII does not reverse the binding [127]. The presence of a competing peptide that blocks the interaction of CaMKII with presynaptic Ca_V_2.1 channels of SCG neurons prevented both PPD and PPF, suggesting that binding of CaMKII to Ca_V_2.1 channels is required for the expression of this regulatory effect. Similarly, the expression of the brain-specific CaMKII inhibitor CaMKIIN [128], which prevents CaMKII binding to Ca_V_2.1 channels [126], also prevented PPD and PPF. Thus, the noncatalytic regulation of Ca_V_2.1 channels by bound CaMKII controls the activity of those channels that have the effector of the Ca^2+^ signal (i.e., CaMKII) in position to bind the entering Ca^2+^ and respond to it [126]. SNARE proteins and RIM similarly serve as effectors of the Ca^2+^ signal for initiation of SVs exocytosis increasing the activity of the Ca_V_2.1 channels by the formation of a complete SNAREs complex with synaptotagmin and RIM bound [53,70]. This “effector checkpoint” mechanism serves to focus Ca^2+^ entry through those Ca^2+^ channels whose effectors are bound and ready to respond.

Furthermore, autophosphorylated CaMKII bound to Ca_V_2.1 channels also binds to synapsin-1, a phosphoprotein of the SVs, increases its phosphorylation and induces oligomers of synapsin-1 [127]. Synapsin-1 is a major presynaptic phosphoprotein that is a prominent substrate for CaMKII, and phosphorylation by CaMKII regulates the effects of synapsin-1 on the trafficking of SVs [129]. The phosphorylation of synapsin-1 by CaMKII increases synaptic transmission at the squid giant synapse [122,123]. Formation of the ternary complex of Ca_V_2.1 and synapsin-1 bound to CaMKII would modulate the dynamics of SV function in AZs containing these proteins [127].

## 7. Neuronal Firing and Presynaptic Short-Term Plasticity

Neuronal firing regulates presynaptic Ca^2+^ channels by Ca^2+^ bound CaM and nCaS and causes facilitation and inactivation of neurotransmitter release. The differential expression of these Ca^2+^-dependent regulatory proteins may provide a means of cell-type-specific regulation of presynaptic Ca^2+^ channels and short-term synaptic plasticity. The short-term plasticity of neurotransmitter release shapes the postsynaptic response to bursts of impulses and is crucial for the fine-grained encoding of information in the nervous system [117,130].

### 7.1. Presynaptic Short-Term Facilitation

The Calyx of Held, the large presynaptic terminal enabling to record directly presynaptic Ca^2+^ current by voltage-clamp methods, suggests that neuronal firing controls P/Q- and N-type currents to modulate differentially synaptic transmission. Presynaptic Ca^2+^ current consists of a combination of P/Q- and N-type currents in young mice and shows activity-dependent facilitation that predicts the amount of synaptic facilitation according to the power law [131,132]. Ca_V_2.1 knockout lost both facilitation of the presynaptic Ca^2+^ current and synaptic facilitation [101,131,132]. The remaining N-type Ca^2+^ currents are less efficient in mediating synaptic transmission and do not support facilitation of synaptic transmission, but they are more sensitive to modulation by G protein-coupled receptors [101]. These results suggest that Ca_V_2.1 channels are responsible for neuronal activity-dependent synaptic facilitation, while Ca_V_2.2 channels have strong G protein regulation.

Presynaptic short APs bursts generate augmentation and longer APs bursts generate post-tetanic potentiation (PTP) relying on residual Ca^2+^. The optical measurement of presynaptic Ca^2+^ transients with the induction of PTP in the calyx of Held showed an increase in the Ca^2+^ influx to the extent that predicted PTP when the power law of neurotransmission was applied, and the Ca^2+^ transient decayed with a time course of the decay of PTP [133]. In Ca_V_2.1-transfected SCG neurons, PTP was not significantly affected by mutations at the IQ-like motif [90]. In contrast, PPF and augmentation share a common mechanism involving an increase in instantaneous Ca^2+^ entry through Ca_V_2.1 channels by CaM- and nCaS-binding in an activity-dependent manner, which in turn facilitates neurotransmitter release. It is likely that facilitation of presynaptic Ca^2+^ currents may contribute to short-term facilitation [90,132], and the augmentation and the PTP represent overlapping processes caused by differential combinations of mechanisms at different synapses [130].

The expression of Ca_V_β subunits has a strong influence on synaptic facilitation in hippocampal synapses through their effects on Ca^2+^ channel function [134]. Cavβ2 and Cavβ4 subunits distribute in clusters and localize to synapses. Caβ2 induces depression, whereas Cavβ4 induces PPF followed by synaptic depression during longer stimuli trains. The induction of PPF by Cavβ4 correlates with a reduction in the release probability and cooperativity of the transmitter release. These results suggest that Cavβ subunits determine the gating properties of the presynaptic Ca^2+^ channels within the presynaptic terminal in a subunit-specific manner and may be involved in the organization of the Ca^2+^ channel relative to the release machinery [134].

The mutation of Ca_V_2.1 channels at the IQ-like motif in hippocampal neurons confirmed the mechanism of short-term synaptic facilitation dependent nCaS regulation of Ca_V_2.1 channels with brief and local Ca^2+^ elevation [135]. In addition, long-term potentiation of synaptic transmission at the Schaffer collateral-CA1 synapse, that is thought to be primarily generated postsynaptically, is substantially weakened by the mutation. Furthermore, the impairments in short-term and long-term plasticity due to Ca_V_2.1 channel mutation at the IQ-like motif are associated with pronounced deficits in spatial learning and memory in context-dependent fear conditioning and in the Barnes circular maze. Thus, regulation of Ca_V_2.1 channels by CaM and nCaS is required for not only presynaptic facilitation but also induction of postsynaptic long-term potentiation, and spatial learning and memory [136].

### 7.2. Presynaptic Short-Term Depression

At the calyx of Held, presynaptic stimulation at 100 Hz induces robust synaptic depression [88]. Synaptic depression during high-frequency APs bursts in presynaptic neurons is generally thought to be a result of SVs depletion [130]. In a prominent feature of synaptic transmission, the depression is caused by a decrease in release probability [103]. The release probability is determined by docked SVs and Ca^2+^ current in the AZ. Presynaptic loading of peptides that disrupt CaM interactions reduced both Ca^2+^-dependent inactivation of the P/Q-type Ca^2+^ current and PPD [88]. The Ca^2+^-dependent inactivation of the presynaptic Ca^2+^ current, rather than SVs depletion, causes rapid synaptic depression for stimuli ranging from 2 to 30 Hz [87,88].

The transfection of SCG neurons with Ca_V_2.1 channels lacking the CBD, a mutation reducing Ca^2+^-dependent inactivation in heterologous expression systems [80,81], blocked PPD, and reduced synaptic depression during APs burst up to 40 Hz [90]. CaBP1 expression, which blocks Ca^2+^-dependent facilitation of P/Q-type Ca^2+^ current, induced PPD, and synaptic depression during APs burst. However, the synaptic depression was absent in the presynaptic neuron coexpressed with CaBP1 and Ca_V_2.1 channels lacking the CBD [113]. These results further demonstrate that rapid synaptic depression is caused by inactivation of presynaptic Ca_V_2.1 channel bound with CaM or CaBP1. During APs burst at 30 Hz and 40 Hz, a slower phase of synaptic depression is likely caused by SVs depletion.

Data from the calyx of Held and Ca_V_2.1-transfected SCG neurons suggest a conserved mechanism for generating rapid synaptic depression evoked by physiological rate and duration (at 40 Hz for 1 s) of APs bursts in multiple synapses where neuronal activity elevates presynaptic Ca^2+^ transient, and such a Ca^2+^ rise dependent binding of nCaS to Ca_V_2 channels inactivates presynaptic Ca^2+^ channels. Studies of β subunits within cultured hippocampal neurons also support an important role for Ca_V_2 channels modulation in synaptic plasticity: the overexpression of Ca_V_β4 favors facilitation whereas the overexpression of Ca_V_β2 favors depression [134].

### 7.3. CaMKII Regulates Short-Term Synaptic Plasticity

The binding of CaMKII to Ca_V_2.1 channels enhances their functional activity by inhibiting their inactivation [126] and enhances the activity of CaMKII by increasing its autophosphorylation [127]. SCG neurons introduced a competing peptide that blocks the interaction of CaMKII with Ca_V_2.1 channels or SCG neurons transfected the brain-specific CaMKII inhibitor CaMKIIN [128] which prevents CaMKII binding to Ca_V_2.1 channels [126] prevented not only PPF and PPD but also synaptic depression during APs burst and augmentation after a conditioning APs burst. It is unlikely that the basal release probability is affected by competing for peptide injection or CaMKIIN expression because the mean amplitudes of the first EPSPs are unchanged. Binding of CaMKII to the Ca_V_2.1 channel is required for both up-regulation of channel activity in presynaptic facilitation and for Ca^2+^-independent activation of CaMKII by Ca_V_2.1, and one or both of these effects is necessary for normal short-term synaptic plasticity.

### 7.4. Ca^2+^-Binding Molecules Regulate Short-Term Synaptic Plasticity

Synaptotagmin-1, 2, and 9 serve as Ca^2+^ sensors to mediate the fast synchronous transmitter release as discussed above [56,73,74]. In contrast, synaptotagmin-7 that binds slowly to Ca^2+^ via its C_2_A domain [137] is not required for the synchronous synaptic transmission but mediates asynchronous transmitter release [111]. Synaptotagmin-7 is also required for the short-term facilitation, such as PPF and synaptic facilitation during APs burst, at several synapses [138]. Synaptotagmin-7 has a stronger contribution to membrane binding, and perhaps to bridging the vesicle and plasma membranes [111] that may enhance the fast transmitter release in response to repetitive APs firing.

In the presynaptic terminal Ca^2+^ buffers such as parvalbumin, calbindin, and related Ca^2+^-binding proteins control Ca^2+^ homeostasis [139] and synaptic strength [140,141,142]. A slow Ca^2+^ buffer parvalbumin [143] controls decay rate of short-term plasticity [144]. In contrast, a rapid Ca^2+^ buffer calbindin [145] alters short-term synaptic facilitation in multiple ways at different synapses [146].

Short-term plasticity may be a combination of the three molecular mechanisms, Ca^2+^ channel modulation, synaptotagmin-7 action and Ca^2+^ buffering, activated by Ca^2+^ elevation with neuronal firing. Ca^2+^ channel modulation with CaM and nCaS is a response to millisecond Ca^2+^ dynamics. The slower synaptotagmin-7 action integrates local and global Ca^2+^ entry, and Ca^2+^ buffering may control the spread Ca^2+^ accumulation [146].

## 8. Conclusions

In response to presynaptic AP firing, Ca^2+^ binding proteins triggers SVs exocytosis and regulate the probability. Thus, modulation of presynaptic Ca^2+^ channels has a powerful influence on synaptic transmission. At the pre-firing state, Ca^2+^ channels activity is inhibited by interaction with AZ proteins. AP firing relieves the inhibition by switching to interact with SNAREs and synaptotagmin, the effectors for Ca^2+^-dependent exocytosis. During and post firing, the activity of the Ca_V_2.1 channel is regulated by interaction with CaM and nCaS dependent on individual speed and affinity of binding to residual Ca^2+^. Interacting with CaMKII, the Ca_V_2.1 channel increases the binding to CaM and nCaS and their interaction causes short-term facilitation and depression of synaptic transmission. Fine-tuning the function and regulatory properties of presynaptic P/Q-type Ca^2+^ currents allow a greater range of input-output relationships and short-term plasticity. In contrast, tonic inhibition of N-type Ca^2+^ currents is activated by G-protein coupled-autoreceptors and retrograde signaling receptors.

## Figures and Tables

**Figure 1 ijms-20-02217-f001:**
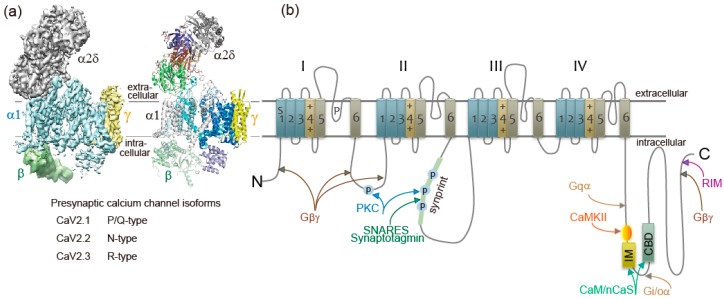
Ca^2+^ channel structure and organization. (**a**) The subunit composition and structure of high-voltage-activated Ca^2+^ channels. The cryo-EM structure of the rabbit voltage-gated Ca^2+^ channel Cav1.1 complex at a nominal resolution of 4.2 Å. The overall EM density map on the left is colored according to different subunits. The structure model on the right is color-coded for distinct subunits. Reproduced from [12]. (**b**) The α1 subunit consists of four homologous domains (I-IV), each consisting of six transmembrane segments (S1-S6). S1–S4 represents the voltage-sensing module. S5–S6 represents the pore-forming unit. The large intracellular loops linking the different domains of the α1 subunit serve as sites of interaction of different regulatory proteins important for channel regulation, including G-protein (Gβγ, Gα), RIM, SNARE proteins, and synaptotagmin at the synprint site (shown in green bar), calmodulin (CaM), and neuronal Ca^2+^ sensor proteins (nCaS) at the IQ-like motif, which begins with the sequence isoleucine-methionine (IM) instead of isoleucine-glutamine (IQ) and the nearby downstream CaM-binding domain (CBD), calmodulin kinase II (CaMKII), and protein kinase C (PKC). Adapted from [4].

**Figure 2 ijms-20-02217-f002:**
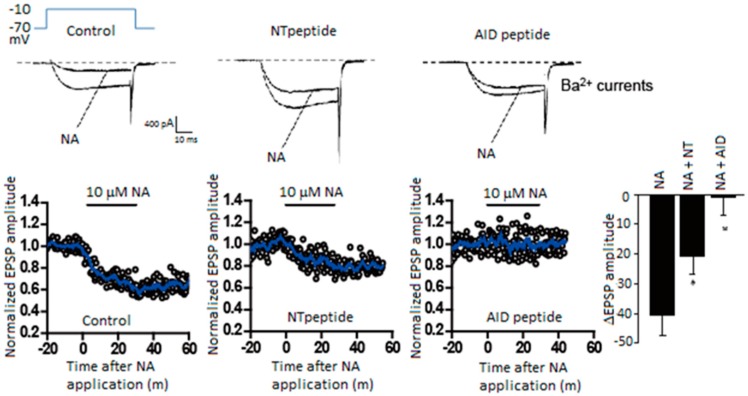
Gβγ-mediated noradrenaline inhibition of transmitter release and N-terminal/I-II loop AID peptides of Ca_V_2.2 α1-subunit. Noradrenaline (NA) induced Ba^2+^ current inhibition (upper traces) and transmitter release (lower graphs) were attenuated in the presence of Gβγ-interaction site of N-terminal peptide (Ca_V_2.2^45-55^, YKQSIAQRART) or AID peptide (Ca_V_^377-393^, RQQQIEREL NGYLEWIF) (See Figure 1b). Ba^2+^ currents were recorded from superior cervical ganglion (SCG) neurons acutely dissociated from 3- to 6-week-old Wistar rats, while the synaptic transmission was recorded from long-term cultured SCG neurons isolated from p7 rat. NA was bath-applied 30 min after injection of the peptide at 1 mM in the injection pipette. EPSPs were normalized to amplitude prior to NA application at time = 0 min. Bar graph summarizing NA effects, **p* < 0.05 *vs.* NA effects in controls (Student’s *t*-test). Adapted from [98].

**Figure 3 ijms-20-02217-f003:**
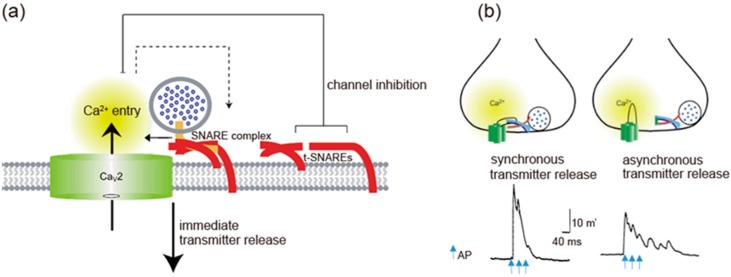
Spatial regulation of transmitter release by the I-II loop interaction with SNAREs. (**a**) The I-II loop interacts with t-SNAREs, resulting in inhibition of Ca2.2 channels opening. Once AP opens the channels, an increase in Ca^2+^ mediates interaction with SNAREs complex and induces transmitter release. Adapted from [4]. (**b**) Triple APs induces a large synchronous transmitter release from the first AP. In contrast, asynchronous transmitter release was observed in the presence of 130 μM synprint peptide (see Figure 1b). Adapted from [99].

**Figure 4 ijms-20-02217-f004:**
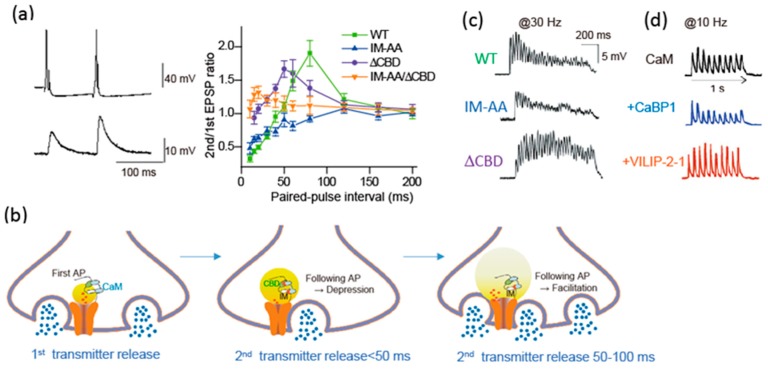
Temporal regulation of Ca^2+^ channel activity by CaM and nCaS after AP(s) firing modulates synaptic transmission. (**a**) Regulation of transmitter release (lower trace) after an AP firing (upper trace). Dependent on the inter-stimulus interval the second AP induces paired-pulse depression (PPD) and facilitation (PPF). The PPD was prevented by ΔCBD, while PPF was prevented by IM-AA mutation of Ca_V_2.1 channels. (**b**) Model for Ca^2+^/CaM-dependent inactivation and facilitation of Ca^2+^ channels and neurotransmitter release. (**c**) Biphasic synaptic transmission during 1-s train of APs at 30 Hz changed to synaptic depression by the IM-AA mutation or to synaptic facilitation by the ΔCBD. (**d**) Overexpression of CaBP1 (blue) blocks synaptic facilitation, while overexpression of VILIP-2 (red) blocks synaptic depression, during 1-s train of APs at 10 Hz. Adapted from [90] (**a**–**c**) and [113] (**d**).

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
