# Peer review of "Presynaptic Calcium Channels"

_ijms, 2019, doi:10.3390/ijms20092217_

Round 1

Reviewer 1 Report

This is a fine review of the topic.

Author Response

Reviewer 1

This is a fine review of the topic.

English language and style are fine/minor spell check required.

Spell check will be done after reviewing my responses to the Reviewer 1 and 2. 

Reviewer 2 Report

This comprehensive review of presynaptic calcium channels and their modulation by various proteins within the nerve terminal could be a welcome addition to the literature and provide a nice perspective on the field by an investigator that has contributed many seminal contributions.  However, in its current form, there are many issues that require attention.

1. The review requires a significant amount of editing for grammar and clarity.  Although the author has contributed many important, clearly written papers to this field in the past, the writing in this review is still very rough in many places.  For example, there are numerous instances of incorrect usage of plurals, incorrect grammar, and sentence construction that makes it difficult to determine the meaning. In all, there are too many of these cases (30-40) to list. 

2. The paragraph on page 2, lines 19-27, discusses the gamma subunit as a member of the presynaptic calcium channel complex, but I am not sure this is correct.  Some have argued that neuronal calcium channels do not contain the gamma subunit.  See for example Muller  et al. (2010), Quantitative proteomics of the Cav2 channel nano-environments in the mammalian brain. Proc Natl Acad. Sci USA 107:14950. Gamma may only be incorporated into skeletal muscle voltage-gated calcium channels. This view has been presented in other reviews of calcium channels and should be discussed here as well.

3. For paragraph 2 line 24-25, the following statement: “The α2δ subunits also set presynaptic release probability [16]” needs to be clarified.  How do they alter synapse function? Are effects on release probability due to alterations in channel function or alpha2-delta effects on other proteins independent of Cav channels (since this auxiliary subunit can function independent of calcium channels)?

3. For figure 1, several edits are suggested:

a) The labels for "extracellular" and "intracellular" are too small to read - in (a) and (b).

b) The segment labels (numbers, +, S) are too lightly colored to read.

c) The subunit labels in (a) are too small.

d) The text for intracellular loop labels should all be the same size and font.

e) In the legend, it is not clear what “domain-colored” means

f) The graphic representation of calcium in the figure: “putative Ca2+ is shown as a green sphere” is not visible.

g) The “sticks” are not easily visible: “The glycosyl moieties are shown as sticks”

4. On page 3, lines 6-7, the statement: “Acetylcholine release from rat sympathetic neurons via presynaptic muscarinic receptors is reduced through this pathway [30]” implies that ACh release is caused by muscarinic receptors. I don't think that is the intention.

5. On page 4, lines 36-38, it would be useful to comment on the mechanism of RGS effects: to speed GTPase activity of the alpha subunit of the activated G protein alpha-GTP.

6. On page 5, line 22, the reference for this statement is incorrect, and I think should be #95?  “Similarly, synprint peptides reduced transmitter release from embryonic Xenopus spinal neurons [57].”

7. On page 6, line 13-28, the paragraph is written as if this is the primary report of this experiment.  It should be re-worded as a summary of previously published data.

8. For the discussion of short-term synaptic facilitation that starts on the bottom of page 6 and recurs in several additional pages (i.e. page 9, line 10), The presentation of CAM-dependent regulation is certainly supported by data in the experimental preparations presented, but there are other mechanisms of short-term presynaptic facilitation that do not involve CaM-dependent regulation.  These alternatives should also be discussed.

9. On page 8, line 38, the presentation of the role of CamKII in vesicle trafficking should be discussed more completely.  As it is presented now, it is not easy to understand. 

10. On page 9, lines 23-24, this sentence: “Expression of CaV subunits has a strong influence on synaptic facilitation in hippocampal synapses through their effects on Ca2+ channel function [118].” seems like an add-on and doesn't flow with the logic of the paragraph.  Perhaps it could be explained with additional text. 

11. For the last sentence on page 9, the statement about the mechanisms that underlie short-term synaptic depression is too restrictive in its focus on the CBD.  This conclusion applies to the preparations cited, but it should be made clear that these conclusions are not generally applicable to all synapses.  Specifically, the role of the CBD is certainly clear, but other mechanisms contribute at some synapses.

12. Section 8: “An Ideal Synapse for studying CaV2.1 and 2.2 Modulation” is not written in a way that integrates well into the review.  This section either requires significant editing or should be deleted.

Author Response

Manuscript ID: ijms-483039
"
Presynaptic Calcium Channels"

Reviewers comments are Arial font. Description in Century font is my replay and Palatino Linotype denote revisions of the text.

Reviewer 2 
Comments and Suggestions for Authors

1. The review requires a significant amount of editing for grammar and clarity.  Although the author has contributed many important, clearly written papers to this field in the past, the writing in this review is still very rough in many places. For example, there are numerous instances of incorrect usage of plurals, incorrect grammar, and sentence construction that makes it difficult to determine the meaning. In all, there are too many of these cases (30-40) to list. 

The text will be English editing after reviewing my revisions.

2. The paragraph on page 2, lines 19-27, discusses the gamma subunit as a member of the presynaptic calcium channel complex, but I am not sure this is correct.  Some have argued that neuronal calcium channels do not contain the gamma subunit.  See for example Muller et al. (2010), Quantitative proteomics of the Cav2 channel nano-environments in the mammalian brain. Proc Natl Acad. Sci USA 107:14950. Gamma may only be incorporated into skeletal muscle voltage-gated calcium channels. This view has been presented in other reviews of calcium channels and should be discussed here as well.

Text was changed as below: Text colored with blue was added.

CaV channels are complexes of a pore-forming a1 subunit and auxiliary subunits. Skeletal muscle CaV channels have four distinct auxiliary protein subunits [8] (Fig. 1a), the intracellular b subunit, the disulfide-linked a2δ subunit complex, and the γ subunit having four transmembrane segments. In contrast, brain neuron CaV2 channels are composed of pore-forming a1 and auxiliary β subunit [14].

3. For paragraph 2 line 24-25, the following statement: “The α2δ subunits also set presynaptic release probability [16]” needs to be clarified.  How do they alter synapse function? Are effects on release probability due to alterations in channel function or alpha2-delta effects on other proteins independent of Cav channels (since this auxiliary subunit can function independent of calcium channels)?

Text was changed as below: Text colored with blue was added.

The α2δ subunits are potent modulators of synaptic transmission. The a2δ subunits increase not only Cav1.2 but also Cav2.2, Cav2.1 currents, suggesting that the α2δ subunits enhance trafficking of the CaV channel complex [17]. Expression of α2δ subunits, that configure presynaptic Ca2+ channels to drive exocytosis through its an extracellular metal ion-dependent adhesion site, sets release probability [18].

3. For figure 1, several edits are suggested:

a) The labels for "extracellular" and "intracellular" are too small to read - in (a) and (b).

b) The segment labels (numbers, +, S) are too lightly colored to read. 

c) The subunit labels in (a) are too small.  d) The text for intracellular loop labels should all be the same size and font.

e) In the legend, it is not clear what “domain-colored” means

f) The graphic representation of calcium in the figure: “putative Ca2+ is shown as a green sphere” is not visible.

g) The “sticks” are not easily visible: “The glycosyl moieties are shown as sticks”

 a)-d) were changed as reviewer’s suggestion.

e) The structure model on the right is color-coded for distinct subunits.

f), g) removed from the legend.

4. On page 3, lines 6-7, the statement: “Acetylcholine release from rat sympathetic neurons via presynaptic muscarinic receptors is reduced through this pathway [30]” implies that ACh release is caused by muscarinic receptors. I don't think that is the intention.

Text was changed as below and text colored with blue was added.

Acetylcholine release from rat sympathetic neurons is reduced through this pathway via presynaptic muscarinic receptors activation

5. On page 4, lines 36-38, it would be useful to comment on the mechanism of RGS effects: to speed GTPase activity of the alpha subunit of the activated G protein alpha-GTP.

Text was changed as below: Text colored with blue was added. 

Regulator of G protein signaling-2 (RGS-2), that speeds GTPase activity of the a subunit of the activated G protein a-GTP, determines short-term plasticity in hippocampal neurons by regulating Gi/o-mediated inhibition of presynaptic Ca2+ channels.

6. On page 5, line 22, the reference for this statement is incorrect, and I think should be #95?  “Similarly, synprint peptides reduced transmitter release from embryonic Xenopus spinal neurons [57].”

New reference [100] was added.

100. Rettig, J., et al., Alteration of Ca2+ dependence of neurotransmitter release by disruption of Ca2+ channel/syntaxin interaction. J Neurosci, 1997. 17(17): p. 6647-56.

7. On page 6, line 13-28, the paragraph is written as if this is the primary report of this experiment.  It should be re-worded as a summary of previously published data.

Text was changed as below:

Modulation of presynaptic Ca2+ channels has a powerful influence on synaptic transmission [90]. The cytoplasmic regions α1 subunit are the target of regulatory proteins for channel modulation (Fig. 1B). Brain-derived α1 subunit of the CaV2.1 channel mediates transmitter release from the transfected SCG neurons [137]. The transmitter release changes after AP firing due to modulation of CaV2.1 channel interacting with Ca2+ bound CaM (Fig. 4) [90]. CaM has two Ca2+ binding sites, N and C robes. The N-robe sensing rapid and higher increase in Ca2+ concentration [111] initiates synaptic depression, and following facilitation is mediated by the C-robe sensing lower Ca2+ concentration. EPSPs recorded by pairs of APs with varied stimulation intervals show paired-pulse depression (PPD) and facilitation (PPF) (Fig. 4a). PPD with short interval (<50ms) was blocked by deletion of the CBD, while PPF with intermediate interval (50-100 ms) was blocked by mutation of the IQ-like motif. Thus, decline in Ca2+ elevation after the first AP causes temporal regulation of the CaV2.1 channel interacting with CaM, resulting in change in the transmitter release efficacy (Fig. 4b). The time-dependent opposing modulation of the CaV2.1 channel activity may support a stable synaptic transmission.

8. For the discussion of short-term synaptic facilitation that starts on the bottom of page 6 and recurs in several additional pages (i.e. page 9, line 10), The presentation of CAM-dependent regulation is certainly supported by data in the experimental preparations presented, but there are other mechanisms of short-term presynaptic facilitation that do not involve CaM-dependent regulation.  These alternatives should also be discussed.

Recent reports for synaptotagmin-7 and calcium buffers in short-term plasticity were added.  

7.4. Ca2+-binding Molecules Regulate Short-Term Synaptic Plasticity

Synaptotagmin-1, 2, and 9 serve as Ca2+ sensors to mediate the fast synchronous transmitter release as discussed above [56, 73, 74]. In contrast, synaptotagmin-7 that binds slowly to Ca2+ via its C2A domain [136] is not required for the synchronous synaptic transmission but mediates asynchronous transmitter release [137]. Synaptotagmin-7 is also required for the short-term facilitation, such as PPF and synaptic facilitation during APs burst, at several synapses [138]. Synaptotagmin-7 has stronger contribution to membrane binding, and perhaps to bridging the vesicle and plasma membranes [137] that may enhance the fast transmitter release in response to repetitive APs firing

In the presynaptic terminal Ca2+ buffers such as parvalbumin, calbindin and related Ca2+-binding proteins control Ca2+ homeostasis [139] and synaptic strength [140-142]. A slow Ca2+ buffer parvalbumin [143] controls decay rate of short-term plasticity [144]. In contrast, a rapid Ca2+ buffer calbindin [145] alters short-term synaptic facilitation in multiple ways at different synapses [146].

Short-term plasticity may be combination of the three molecular mechanisms, Ca2+ channel modulation, synaptotagmin-7 action and Ca2+ buffering, activated by Ca2+ elevation with neuronal firing. Ca2+ channel modulation with CaM and nCaS is response to millisecond Ca2+ dynamics. The slower synaptotagmin-7 action integrates local and global Ca2+ entry, and Ca2+ buffering may control the spread Ca2+ accumulation [146].

9. On page 8, line 38, the presentation of the role of CamKII in vesicle trafficking should be discussed more completely.  As it is presented now, it is not easy to understand. 

Text was changed as below: Text colored with blue was added.

Furthermore, autophosphorylated CaMKII bound to CaV2.1 channels also binds to synapsin-1, a phosphoprotein of the SVs, increases its phosphorylation and induces oligomers of synapsin-1 [126]. Synapsin-1 is a major presynaptic phosphoprotein that is prominent substrate for CaMKII, and phosphorylation by CaMKII regulates the effects of synapsin-1 on trafficking of SVs [128]. Phosphorylation of synapsin-1 by CaMKII increases synaptic transmission at the squid giant synapse [121, 122]. Formation of the ternary complex of CaV2.1 and synapsin-1 bound to CaMKII would modulate the dynamics of SV function in active zones containing these proteins [126].

10. On page 9, lines 23-24, this sentence: “Expression of CaVb subunits has a strong influence on synaptic facilitation in hippocampal synapses through their effects on Ca2+ channel function [118].” seems like an add-on and doesn't flow with the logic of the paragraph.  Perhaps it could be explained with additional text

Text colored with blue was added:

Expression of CaVb subunits has a strong influence on synaptic facilitation in hippocampal synapses through their effects on Ca2+ channel function [133]. Cavb2 and Cavb4 subunits distribute in clusters and localize to synapses. Cab2 induces depression, whereas Cavb4 induces PPF followed by synaptic depression during longer stimuli trains. The induction of PPF by Cavb4 correlates with a reduction in the release probability and cooperativity of the transmitter release. These results suggest that Cavb subunits determine the gating properties of the presynaptic Ca2+ channels within the presynaptic terminal in a subunit-specific manner and may be involved in organization of the Ca2+ channel relative to the release machinery [133]. 

11. For the last sentence on page 9, the statement about the mechanisms that underlie short-term synaptic depression is too restrictive in its focus on the CBD.  This conclusion applies to the preparations cited, but it should be made clear that these conclusions are not generally applicable to all synapses.  Specifically, the role of the CBD is certainly clear, but other mechanisms contribute at some synapses.

Text colored with blue was added:

 7.4. Ca2+-binding Molecules Regulate Short-Term Synaptic Plasticity

Synaptotagmin-1, 2, and 9 serve as Ca2+ sensors to mediate the fast synchronous transmitter release as discussed above [47, 64, 65]. In contrast, synaptotagmin-7 that binds slowly to Ca2+ via its C2A domain [124] is not required for the synchronous synaptic transmission but mediates asynchronous transmitter release [125]. Synaptotagmin-7 is also required for the short-term facilitation, such as PPF and synaptic facilitation during APs burst, at several synapses [126]. Synaptotagmin-7 has stronger contribution to membrane binding, and perhaps to bridging the vesicle and plasma membranes [125] that may enhance the fast transmitter release in response to repetitive APs firing

In the presynaptic terminal Ca2+ buffers such as parvalbumin, calbindin and related Ca2+-binding proteins control Ca2+ homeostasis [127] and synaptic strength [128-130]. A slow Ca2+ buffer parvalbumin [131] controls decay rate of short-term plasticity [132]. In contrast, a rapid Ca2+ buffer calbindin [133] alters short-term synaptic facilitation in multiple ways at different synapses [134].

Short-term plasticity may be combination of the three molecular mechanisms, Ca2+ channel modulation, synaptotagmin-7 action and Ca2+ buffering, activated by Ca2+ elevation with neuronal firing. Ca2+ channel modulation with CaM and nCaS is response to millisecond Ca2+ dynamics. The slower synaptotagmin-7 action integrates local and global Ca2+ entry, and Ca2+ buffering may control the spread Ca2+ accumulation [134].

12. Section 8: “An Ideal Synapse for studying CaV2.1 and 2.2 Modulation” is not written in a way that integrates well into the review.  This section either requires significant editing or should be deleted.

Text Section 8 was deleted, and introduction of Section 6 was added.

At most fast synapse in the central nervous system, CaV2 channels are expressed diversely. In contrast, synaptic transmission of long-term cultured sympathetic SCG neurons, forming a well-characterized cholinergic synapse [107, 108], is mediated by CaV2.2 channels [109, 110]. Physiological role of presynaptic CaV2.1 channel modulation by Ca2+-sensors was explored by exogenously expressed a1 subunit derived from the brain CaV2.1 channel that functionally generate P/Q type currents with other endogenous subunits in SCG neuron [137]. This section 6 describes presynaptic plasticity induced by modulation of the CaV2.1 channel that is mediated by CaM or expression of neuron specific Ca2+-sensor proteins, monitoring excitatory postsynaptic potentials (EPSPs) evoked by various pattern of presynaptic APs firing in the presence of the blocker of endogenous CaV2.2 channels [109].

Reviewer 3 Report

This is an authoritative review of the role and modulation of calcium channels in synaptic transmission.  The review is balanced and accurate.  A bit more detail could have been provided on G protein modulation of the channels, such as Gbeta subtype specificity and the effect of different calcium channel beta subunits, along with the interplay between syntaxin and G lrotein modulation.  Otherwise I like it!  

There are a number of small English mistakes such as the incorrect use of plural with “channels” - this needs to be fixed

Author Response

Manuscript ID: ijms-483039
"
Presynaptic Calcium Channels"

Reviewers comments are Arial font. Description in Century font is my replay and Palatino Linotype denote revisions of the text.

Reviewer 3

This is an authoritative review of the role and modulation of calcium channels in synaptic transmission.  The review is balanced and accurate.  A bit more detail could have been provided on G protein modulation of the channels, such as Gbeta subtype specificity and the effect of different calcium channel beta subunits, along with the interplay between syntaxin and G protein modulation.  Otherwise I like it!  

Description for Gbeta subtype specificity, Effect of different calcium channel beta subunits and Interplay between syntaxin and G protein modulation was added.

Gbeta subtype specificity:

Specific Gβ subunit is responsible for the CaV2.2 channel modulation in different neurons. In rat SCG neurons CaV2.2 channels are differentially modulated by different types of Gβ subunits, with Gβ1 and Gβ2 being most effective, Gβ5 showing weaker modulation, and Gβ3 and Gβ4 being ineffective [32-34]. In contrast, in rat stellate ganglia neurons, 2 and Gβ4 but not 1 subunit are responsible for the coupling of CaV2.2 channels with noradrenaline receptors [35]. In the transfected human embryonic kidney tsa201 cell line, CaV2.2 channel inhibition, with Gβ1 and Gβ3 being more effective than Gβ4 and Gβ2, and no significant modulation being induced by Gβ5 [36]. Gβ subunitinduced inhibition of CaV2.1 channels differed from those observed with the CaV2.2 channel. CaV2.1 channels exhibited more rapid rates of recovery from inhibition than those observed with CaV2.2 channels, on average, twice as rapidly for the CaV2.1 channels, indicating that Gβ binding to this channel subtype is less stable [36].

Effect of different calcium channel beta subunits:

The subtype of CaVβ can influence the extent and kinetics of Gβγ mediated inhibition and this depends on the subtype of Gβ involved [37, 38]. Gβγ interacts with multiple sites on the N-terminus, I–II linker, and the C-terminus of the a1 subunit. Binding of Gβγ causes a conformational shift that promotes interaction of the N-terminus “inhibitory module” with the initial one-third of the I–II-linker. Strong membrane depolarization leads to unbinding of Gβγ and loss of interaction between the N-terminus and the I–II linker. This depends upon binding of CaVβ subunit to the AID on the I–II linker. In the absence of CaVβ1 subunit binding with mutation in the AID, Ca2+ channel inhibition still occurs but cannot be reversed by strong depolarization. CaVβ2a, that is palmitoylated at two N-terminal cysteine residues, can still take place and permit voltage-dependent relief of the inhibition [39]. It is possible that binding of CaVβ1 to the AID induces a rigid α-helical link with domain IS6, and this transmits movement of the voltage-sensor and activation gate to the I–II linker to alter the Gβγ binding pocket at depolarized potentials [40].

Interplay between syntaxin and G protein modulation:

Text colored with blue was added:

Regulation of the CaV2.2 channels also involves interplay between Ca2+ channels and G protein interaction. Syntaxin-1A, a presynaptic plasma membrane protein, is required for G protein inhibition of presynaptic Ca2+ channels [41]. Physical interactions between syntaxin-1A and Ca2+ channels is a prerequisite for tonic Gbγ modulation of CaV2.2 channels, suggesting that syntaxin 1A mediates a colocalization of Gβγ subunits and CaV2.2 channels, thus resulting in more effective G protein coupling to, and regulation of, the channel. The interactions between syntaxin, G proteins, and CaV2.2 channels are part of the structural specialization of the presynaptic terminal [42]. The interactions between syntaxin, G proteins, and CaV2.2 channels are part of the structural specialization of the presynaptic terminal [42].

There are a number of small English mistakes such as the incorrect use of plural with “channels” - this needs to be fixed

The text will be English editing after reviewing my revisions.

Round 2

Reviewer 2 Report

I am satisfied with the revisions to the content of the manuscript, and have no further suggestions for content editing. The use of the English language will require significant editing.

This manuscript is a resubmission of an earlier submission. The following is a list of the peer review reports and author responses from that submission.

Round 1

Reviewer 1 Report

This review is nicely done with presenting in new findings and bringing up to speed the reader in the filed related to Ca2+ channels located in the presynaptic terminal of neurons.

It is nicely done in covering different preparations such as brain slices and NMJ. I feel a number of people will cite this article as a review in primary research papers. I did not note any changes needed.